# sRNA-mediated regulation of *gal* mRNA in *E. coli*: Involvement of transcript cleavage by RNase E together with Rho-dependent transcription termination

**Heung Jin Jeon** [1,2]☯, **Yonho Lee** [1]☯, **Monford Paul Abishek N** [1]☯, **Xun Wang** [3], **Dhruba K. Chattoraj** [4], **Heon M. Lim** [1]*

1 Department of Biological Sciences, College of Biological Sciences and Biotechnology, Chungnam National University, Daejeon, Republic of Korea, 2 Infection Control Convergence Research Center, College of Medicine, Chungnam National University, Daejeon, Republic of Korea, 3 State Key Laboratory of Agricultural Microbiology, College of Life Science and Technology, Huazhong Agricultural University, Wuhan, PR China, 4 Basic Research Laboratory, Center for Cancer Research, National Cancer Institute, National Institutes of Health, Bethesda, Maryland, United States of America

☯ These authors contributed equally to this work.
* hmlim@cnu.ac.kr

**Data Availability Statement:** All relevant data are within the manuscript and its Supporting Information files.

## Abstract

In bacteria, small non-coding RNAs (sRNAs) bind to target mRNAs and regulate their translation and/or stability. In the polycistronic *galETKM* operon of *Escherichia coli*, binding of the Spot 42 sRNA to the operon transcript leads to the generation of *galET* mRNA. The mechanism of this regulation has remained unclear. We show that sRNA-mRNA base pairing at the beginning of the *galK* gene leads to both transcription termination and transcript cleavage within *galK*, and generates *galET* mRNAs with two different 3'-OH ends. Transcription termination requires Rho, and transcript cleavage requires the endonuclease RNase E. The sRNA-mRNA base-paired segments required for generating the two *galET* species are different, indicating different sequence requirements for the two events. The use of two targets in an mRNA, each of which causes a different outcome, appears to be a novel mode of action for a sRNA. Considering the prevalence of potential sRNA targets at cistron junctions, the generation of new mRNA species by the mechanisms reported here might be a widespread mode of bacterial gene regulation.

## Author summary

sRNAs are regulators of gene expression in all forms of life. In bacteria such as *E. coli*, sRNAs base-pair with mRNA, which can have many consequences such as premature transcription termination by Rho and degradation of mRNA by the endoribonuclease RNase E. Here we show that the two processes can occur on the same sRNA-mRNA pair and produce shorter but stable mRNA that can be of functional significance. Thus, in sRNA regulation of mRNA, transcript cleavage can be an additional mechanism to well-

**Funding:** This research was funded by the Basic Science Research Program of the National Research Foundation of Korea: 2017R1D1A3B03027965, 2018R1D1A3B07044032, 2020R1A2C200633611 to H.M.L., and 2017R1A5A2015385 to H.J.J. This research was also funded by the Intramural Research Program of the Center for Cancer Research, NCI, NIH, of USA to D.K.C. The funders had no role in study design, data collection and analysis, decision to publish, or preparation of the manuscript.

**Competing interests:** The authors have declared that no competing interests exist.

known transcription termination to generate new mRNA species. The choice between the mechanisms is dictated by where on the mRNA the base pairing with sRNA takes place.

## Introduction

The galactose (*gal*) operon of *Escherichia coli* has been the subject of many seminal studies on transcription regulation in bacteria [1]. Four structural genes comprise this operon: *galE*, *galT*, *galK*, and *galM* (Fig 1A). In addition to the full-length *gal* mRNA, *galETKM* (formerly called mM1), four other mRNA products are also found: *galE*, *galE1*, *galET* and *galETK* [2]. These mRNAs have the same 5' end at the transcription initiation site, but different 3' ends immediately after the *galE*, *galT*, or *galK* genes (yielding *galE*, *galET*, or *galETK* mRNA, respectively). The 3' end of the *galE1* mRNA lies in the middle of the *galT* gene.

The 3' end of the full-length *galETKM* mRNA is located at *gal* operon nucleotide coordinate 4,313, where 1 is the transcription initiation site of the *galP1* promoter (Fig 1A) [2,3]. The 3' end is generated due to Rho-dependent transcription termination (RDT) followed by exonucleolytic processing up to 4,313, the currently accepted mature 3' end of *galETKM* mRNA [4]. Another type of transcription termination, known as intrinsic or Rho-independent transcription termination, generates the 3' end at 4,315, which is immediately processed by RNase II to 4,313 [4]. Rho also causes natural polarity in the *gal* operon, which results in greater levels of promoter-proximal transcripts [5]. The Rho-dependent termination that occurs at the end of each gene has been suggested as the key molecular event underlying the natural polarity in *gal* gene expression [1,2].

Bacteria cope with a rapidly changing environment by regulating gene expression. One such adaptive response is mediated by small RNAs (sRNAs), a group of regulatory RNAs that are encoded *in trans* to the target mRNAs to which they base-pair [6]. Most sRNAs bind near the 5' end of target mRNA and mediate negative control [7]. sRNA binding near the translation initiation site of target mRNA blocks translation initiation. This may also allow recruitment of RNase E, which cleaves in the ribosome-free zones of mRNA that are created as a result of sRNA binding; this augments negative control due to degradation of the target mRNA [7–12].

Another consequence of sRNA binding at the 5' end of target mRNA is Rho-dependent termination (RDT). Transcription without accompanying translation makes the ribosome-free transcript available for Rho binding and subsequent transcription termination [1,13–16]. For example, the sRNA ChiX binds to the 5' end of its target mRNA *chiPQ*, which causes decoupling of translation from transcription, resulting in RDT downstream from the point of decoupling [17–19]. At the intercistronic region between *galT* and *galK*, the stop codon of *galT* (blue colored UAA) and the initiator codon of *galK* (green colored AUG) are separated by 3 nucleotides (Fig 1A). A putative ribosome-binding site for *galK* (red colored GGAG) exists two nucleotides upstream of the *galT* stop codon. The sRNA, Spot 42, is known to be involved in the regulation of this *galT-galK* region [20]. Spot 42 is a 109-nucleotide RNA found in *E. coli* and some 196 other species of Gammaproteobacteria [21]. Spot 42 binds near the *galK* start codon and inhibits the translation of *galK* (Fig 1A) [20,22]. This binding also results in the decrease of *galK* transcription [23]. In a previous study, we reported that Spot 42 decreases *galK*-specific mRNA by accelerating the degradation of an internal product of the *gal* operon, the *galKM* mRNA (formerly called mK2) [24]. However, we also found that Spot 42 binding causes transcription termination at the end of *galT*, generating a new mRNA species, *galET* (formerly called mT1). Thus, by generating the promoter-proximal *galET* mRNA and

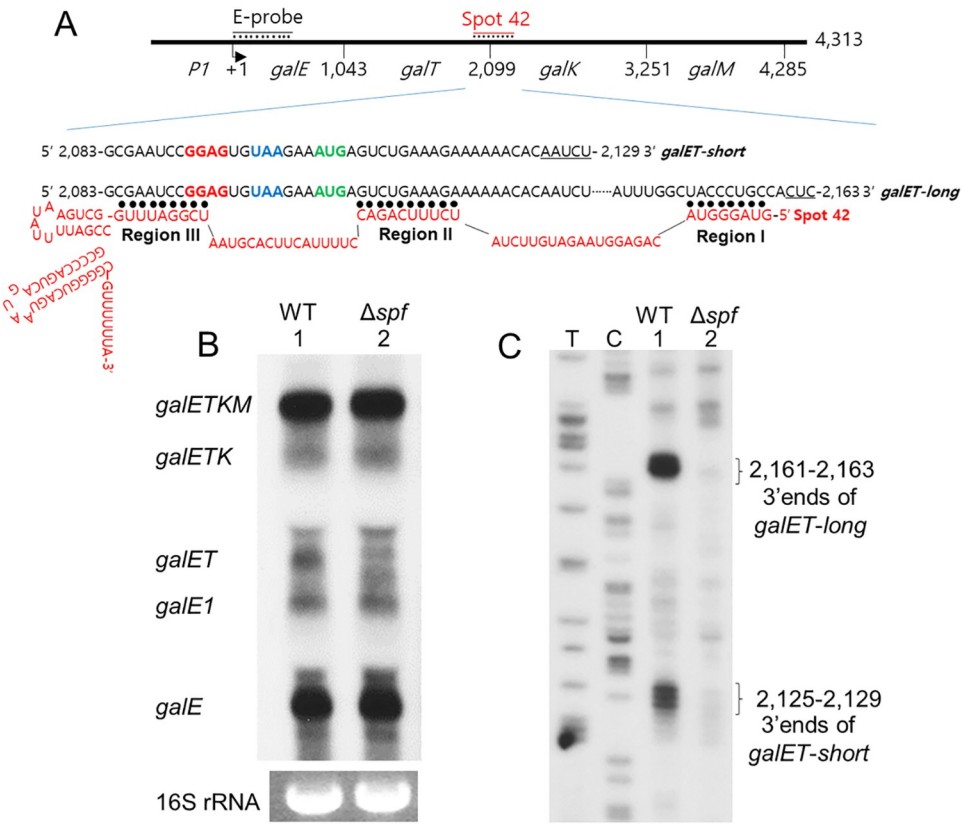

**Fig 1. Spot 42 promotes generation of *galET* mRNA form *galETKM* operon.** A) Schematics of the *galETKM* operon showing the position of the last nucleotide of the stop codon of each gene from the transcription initiation site of the *P1* promoter (+1) (top line). The Spot 42 (red) binding site spans about 75 nucleotides and is located at the *galT-galK* junction [20]. The region of *galE* used to probe northern blots is marked as E-probe. The next two lines show the nucleotide sequence of the 3' ends of *galET-short* and *galET-long* mRNA. The letters in bold are a putative Shine-Dalgarno sequence (GGAG, red letters) for *galK*, the stop codon (UAA, blue letters) of *galT*, and the initiator codon (AUG, green letters) of *galK*. The sequence of the entire Spot 42 RNA is shown in red and its regions that are complementary to *gal* mRNA are indicated by black dots and are marked as Region I, II and III. B) Northern blot with the E probe of *gal* mRNA from MG1655 (WT, lane 1) and MG1655Δ*spf* (Δ*spf*, lane 2) cells. Note that the *galET* mRNA band is missing in lane 2. The identity of the two faint bands in MG1655Δ*spf* cells where the *galET* band resides has not been determined. Shown at the bottom is 16S rRNA, which was used as a loading control. C) 3' RACE assay of 3' ends of *galET* mRNA from MG1655 (lane 1) and MG1655Δ*spf* (lane 2) cells. DNA sequencing ladders that serve as length markers are in lanes marked T and C. The data presented in these blots are representative of three independent experiments.

degrading the promoter-distal *galKM* mRNA, Spot 42 enhances natural polarity in the *gal* operon [24].

In this study, we further explore the role of Spot 42 binding at the *galT-galK* junction in the generation of *galET* mRNA species. We find that the *galET* mRNA has two alternate 3' ends located at positions 30 and 60 nucleotides downstream of the *galT* stop codon (Fig 1A) [24]. We show that these two different 3'ends are generated as a result of base-pairing of two different regions of Spot 42 with the corresponding complementary regions of mRNA and involvement of two different mechanisms: RDT (as was already known) [24] and a previously unreported RNase E-mediated transcript cleavage. We demonstrate that RNase E generates exclusively the longer product and Rho primarily the shorter product after processing by RNase III. Spot 42 binding thus causes generation of different *galET* species by two different 3' end-generating mechanisms. sRNA-control is known to be primarily at the 5' end of messages.

*gal* operon is the first example where sRNA regulation is seen internal to the message. Finally, we discuss the possible physiological relevance of generating variously truncated stable versions of a polycistronic message.

## Results

### Spot 42 promotes processing of *gal* operon mRNA to *galET* mRNA

We performed a northern blot of total RNA prepared from WT (MG1655) cells grown in LB supplemented with 0.5% galactose to an $OD_{600}$ of 0.6, the growth phase up to which Spot 42 production remains maximal [23]. We analyzed the blot with an E-probe that hybridizes to the first half of the first gene of the operon, *galE* (Fig 1A). The results show that the following *gal* specific mRNAs are present in WT cells: *galE*, *galE1*, *galET*, *galETK*, and the full-length *galETKM* (Fig 1B, lane 1). In our previous study, we determined the 3' ends of *gal* mRNAs across the entire operon and found that the 3' ends mentioned in Fig 1B are the major 3' ends produced from the operon [2].

In this study, we measured the location of the 3' ends of the *galET* mRNA more precisely using the 3' RACE assay. The 3'ends of *galET* clustered at two different locations in the *gal* operon between nucleotide coordinates 2,125–29 and 2,161–63 (Fig 1C, lane 1). The two 3' end clusters were located approximately 30 and 60 nucleotides downstream from the stop codon of *galT* (Fig 1A). Since the 5' end of *galET* resides at the transcription initiation site [2], these results indicate that the single mRNA band observed in the northern blot was actually composed of two clusters of mRNA species that differ in their 3' ends.

We found that the *galET* mRNA was specifically reduced to 30 (±3) % of WT in MG1655Δ*spf* cells in which the gene for the Spot 42 sRNA is deleted (Fig 1B, lane 2). All the other mRNA bands in MG1655Δ*spf* showed more or less the same amount of WT (Fig 1B). In such cells, both the 3' end clusters of the *galET* mRNA were missing (Fig 1C, lane 2) (S1 Fig). These results indicate that Spot 42 is required to generate both the clusters. In this study, we referred to the *galET* mRNA that has 3' ends about 60 nucleotides downstream from the *galT* stop codon as *galET-long*, and that has 3' ends about 30 nucleotides downstream from the *galT* stop codon as *galET-short* (Fig 1C).

### Specific base-pairing of Spot 42 and *gal* mRNA is critical for the generation of *galET* 3' ends

i) *Mutations in region I of Spot 42 abrogate the generation of* galET-long *but not* galET-short. sRNAs commonly function through base pairing to target mRNAs. Nucleotide sequence analysis indicated that regions I, II and III of Spot 42, identified by Beisel and Storz [23], could each make eight to ten perfect base pairings with three different regions of the *gal* mRNA (Fig 1A). We investigated the role of each of the base pairing regions of Spot 42 on the generation of the 3' ends of *galET*.

Region I is located at the 5' end of Spot 42 that has a guanine complementary to the cytosine at coordinate 2,158 of *gal* mRNA (Fig 2A). It is likely that region I basepairs with 8 consecutive nucleotide sequences of *gal* mRNA, a few nucleotides upstream from what would be the 3' ends of *galET-long*. We deleted either one or three nucleotides from the 5' end of region I. Using 3'RACE, we assayed the 3' ends in MG1655Δ*spf* cells that harbor a pBR322-derived expression plasmid pSpot42, in which different mutations in the Spot 42 gene (*spf*) were introduced [23]. The 3'RACE results revealed that a single nucleotide deletion in Spot 42 (as in Spot42DMI-1, Fig 2A) resulted in production of the 3' end of *galET-long* about two nucleotides shorter than WT *galET-long* (Fig 2B, lane 2). The deletion of three nucleotides (as in

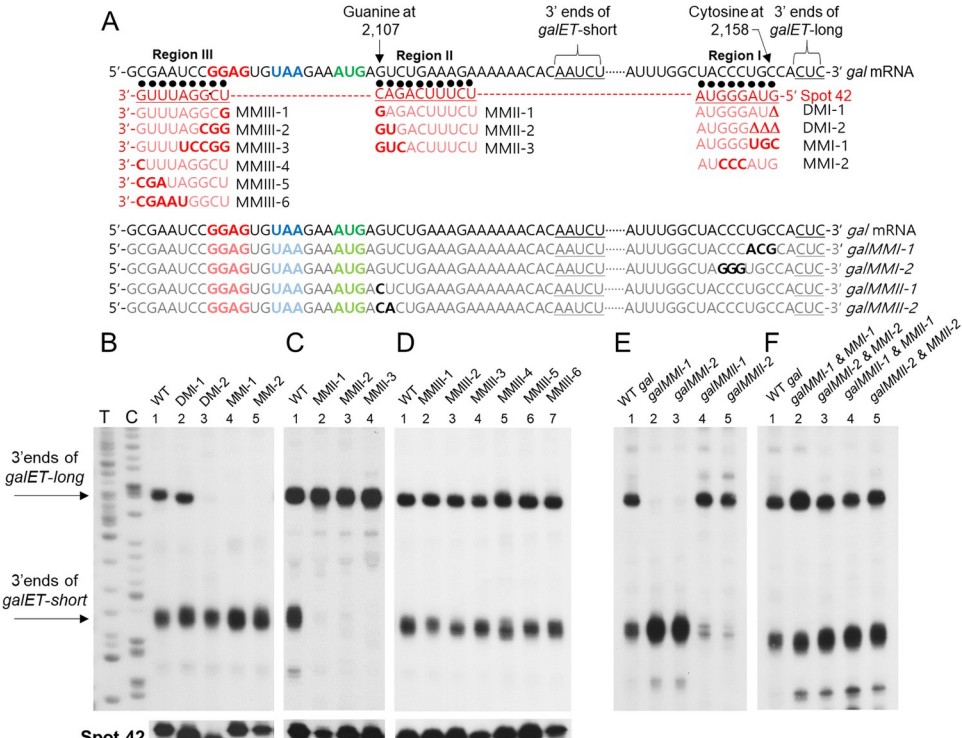

**Fig 2. Specific base-pairing of Spot 42 and *galET* mRNA is critical for the generation of *galET* 3' ends.** A) The top two lines show the sequence stretches of Spot 42 Regions I-III that are complementary to *gal* mRNA as in Fig 1A. The 3' end sequences of *galET-short* at 2,125–9 and *galET-long* at 2,161–3 are indicated by inverted braces as well as underlined. Guanine at 2,107 and Cytosine at 2,158 are indicated to help locate the mutated nucleotides shown in the lines that follow. The base changes are in bold. The upper lines show the changes in Spot 42 and lower lines the changes in *gal* mRNA. B) 3' RACE assay of 3' ends of *galET* RNA from MG1655Δ*spf* cells carrying a plasmid-borne WT Spot 42 gene or its Region I mutant derivatives. Shown at the bottom are 5' RACE assay to indicate the relative amounts of Spot 42 WT and mutant RNAs in different lanes. C) Same as (B) except that Spot 42 was mutated in Region II. D) Same as (B) except that Spot 42 was mutated in Region III. E) 3' RACE assay of 3' ends of *galET* RNA from MG1655Δ*gal* carrying different plasmid-borne *gal* mutant operons. F) 3' RACE assay of 3' ends of *galET* RNA from MG1655Δ*gal*Δ*spf* cells carrying two different plasmids, one with mutated *gal* operon and the other with mutated Spot 42 gene. The plasmid pairs were chosen so that their mutated bases are complementary to each other. The data presented in these blots are representative of three independent experiments.

Spot42DMI-2, Fig 2A) resulted in a drastic reduction in the level of *galET-long* (Fig 2B, lane 3). The level of *galET-short* in the two mutants was essentially the same as it was when Spot 42 was WT (Fig 2B, lane 1). These results indicate that the 5' end of region I of Spot 42 plays an important role in the formation of *galET-long* specifically.

We next introduced substitution mutations in region I of Spot 42. The three nucleotides at the 5' end of region I (5' GUA) were changed to their complementary nucleotides to yield Spot42MMI-1, and the next three nucleotides (5' GGG) were changed to their complementary sequence to generate Spot42MMI-2 (Fig 2A). Results from 3'RACE revealed that the formation of *galET-long* was completely eliminated when Spot 42 had either of the substitution mutations (Fig 2B, lanes 4 and 5). Again, these region I changes had only a marginal effect on the level of *galET-short* (Fig 2B, lanes 1–5). Thus, the results of both the deletion and substitution mutations in region I of Spot42 suggest that pairing of region I with its complementary sequences in *gal* mRNA plays a critical role in the generation of *galET-long* specifically.

ii) *Mutations in region II of Spot 42 abrogate the generation of galET-short but not* galET-long. Spot 42 region II has perfect complementarity to a stretch of 10 contiguous nucleotides

of *gal* mRNA (Fig 2A). The 3' end of region II of Spot 42 is complementary to the second nucleotide following the initiator codon of *galK* (the guanine at 2,107, Fig 2A). We made a series of substitution mutations in region II and assayed the effect of these mutations on the formation of *galET* 3' ends. One, two or three nucleotides from the 3' end of region II were changed to their complementary sequences, generating Spot 42 mutants MMII-1, -2 and -3, respectively (Fig 2A). Results of 3'RACE showed that substituting a single-nucleotide in Spot 42, as in Spot42MMII-1, caused a drastic reduction in the level of *galET-short* (Fig 2C, lane 2). The results were the same when two- and three-nucleotide substitutions were made (Fig 2C, lanes 3 and 4). Generation of *galET-long* was not affected by the presence of these region II mutants (Fig 2C, lanes 1–4), suggesting that the pairing event that generates *galET-long* was not affected by un-pairing of the three nucleotides of Spot 42 region II.

iii) *Mutations in region III of Spot 42 do not affect the generation of either* galET-long *or* galET-short. Region III of Spot 42 has perfect complementarity to 9 consecutive nucleotides partially covering the Shine-Dalgarno sequence of *galK* mRNA (GGAG, Fig 2A). Unlike mutations in regions I and II, none of the substitution mutations in region III of Spot 42 affected generation of either of the 3' ends of *galET* mRNA significantly (Fig 2D, lanes 1–7). We note that when Spot 42 mutants caused drastic changes in the level of *galET* mRNAs, they were not due to drastic changes in Spot 42 mutant RNA levels (Fig 2B–2D, bottom row).

To further examine the specific base-pairing requirements of Spot 42 and *gal* mRNA, we took a complementary approach: Instead of changing Spot 42, we made substitution mutations in some of the *gal* sequences implicated in base-pairing with Spot 42 and assayed the formation of *galET* 3' ends. Mutations were generated in a pBAC-derived low copy number plasmid, pGal that carries the entire *gal* operon [25]. The effect of *gal* mutations was assayed in the MG1655Δ*gal* strain [25].

We substituted 3 nucleotides of pGal, generating *galMMI-1* that would change *gal* mRNA from 5' UGC to 5' ACG, and thus break the complementarity with the AUG at the 5' end of Spot 42 (Fig 2A, lower set of sequence lines). Results of 3' RACE showed that the 3' end of *galET-long* formation was severely impaired (Fig 2E, lane 2). The results were same when we substituted the next 3 nucleotides of *gal*, from CCC to GGG, generating *galMMI-2* (Fig 2A, also see Fig 2E, lane 3). In contrast, neither of the *gal* mutants caused any impairment in the generation of *galET-short*. These results, in conjunction with the results from the Spot 42 region I mutants Spot42MMI-1 and -2 (which have changes in the same base-pairing region as the *gal* mutants *galMMI-1* and *-2*), led us to conclude that specific base-pairing between region I of Spot 42 and *gal* mRNA is critical for the formation of the 3' ends of *galET-long*.

We took a similar approach to test the base-pairing requirements for the generation of *galET-short*. The guanine at 2,107 in *gal* mRNA, which pairs with the 3' end of region II of Spot 42, was changed to cytosine, generating the *gal* mutant *galMMII-1* (Fig 2A, last but one of the sequence lines). Results of 3' RACE showed that the level of the 3' end of *galET-short* decreased to 40 (±3) % of the level observed in the absence of the mutation (Fig 2E, lane 4). When the next base was changed additionally, generating *galMMII-2*, the level of the 3' end of *galET-short* decreased to less than 20 (±4) % of the level observed in the absence of the mutations (Fig 2E, lane 5). Generation of *galET-long* was not affected by these mutations (Fig 2E, lanes 1, 4 and 5). Together with the results of Spot 42 region II mutants Spot42MMII-1 and -2 (which have changes in the same base-pairing regions as the *gal* mutants *galMMII-1* and *-2*), these results led us to conclude that base-pairing between region II of Spot 42 and *gal* mRNA is critical for the formation of 3' ends of *galET-short*.

Next, we tested if these *gal* mutants could restore *galET* production to its WT level in the presence of Spot 42 mutants with complementary changes (Fig 2F). When MG1655Δ*gal*Δ*spf* cells harboring both pGal and pSpot 42 mutant plasmids with complementary changes were

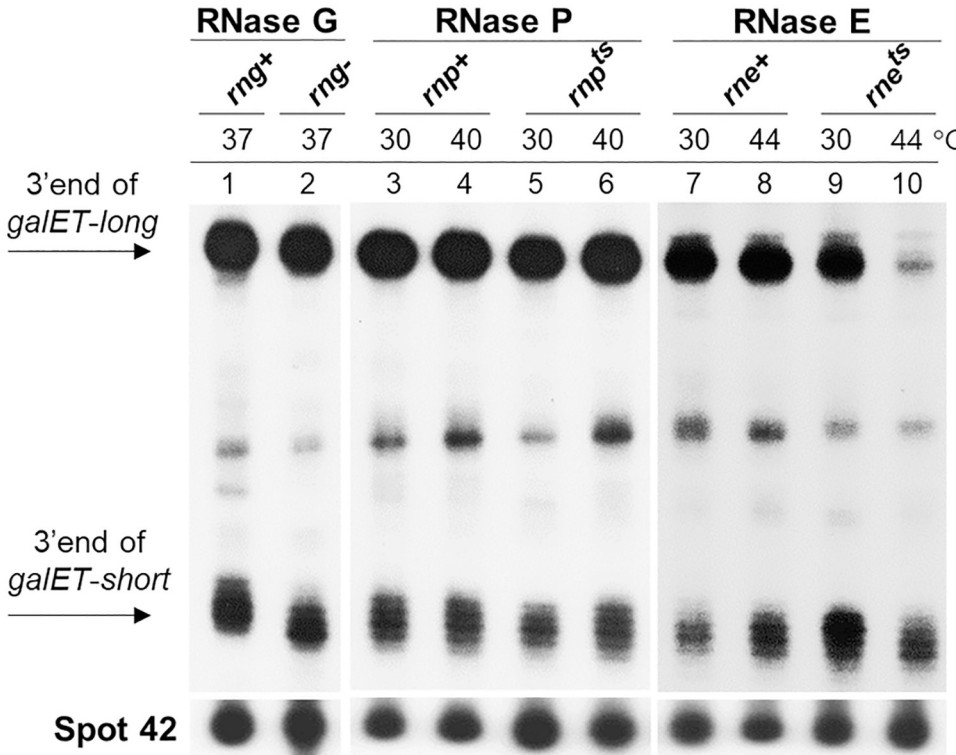

**Fig 3. RNase E is involved in the generation of *galET-long* 3' ends.** 3'RACE assay of *galET* RNA 3' ends in three different endoribonuclease mutant strains. Strains used are GW10 (*rng+ rne+*; WT for RNase G and RNase E, lanes 1, 7 and 8), GW11 (*rng::cat*; [*rng-*] RNase G down mutant, lane 2), NHY312 (*rnpA+*; [*rnp+*] WT for RNase P, lanes 3 and 4), NHY322 (*rnpA49*; [*rnp^ts*] RNase P temperature-sensitive mutant, lanes 5 and 6), GW20 (*ams1^ts*; [*rne^ts*] RNase E temperature-sensitive mutant, lanes 9 and 10). The data presented in these blots are representative of three independent experiments.

assayed for the presence of the 3' ends of the *galET* mRNA, the levels of both of the 3' ends were restored to WT levels in all cases (Fig 2F). These results confirm that base pairing is required to generate both *galET-long* and *galET-short*, and the regions of base-pairing involved in their generation are different.

## RNase E is involved in the generation of *galET-long* 3' ends

In our previous study, we suggested that RDT is the molecular event that generates the 3' end of *galET* mRNA [24]. Because we found that there are two different 3' ends of *galET* mRNA, we tested the possibility that endoribonuclease-mediated transcript cleavage could be another mechanism to generate one of the 3' ends. We assayed the 3' ends of *galET* mRNA in strains defective in endoribonucleases RNase G (*rng^-*), RNase P (*rnp^ts*) and RNase E (*rne^ts*). In *rng^-* [26], both *galET-long* and *galET-short* were produced in amounts comparable to that in WT, indicating that RNase G is not involved in the generation of the two *galET* species (Fig 3A, lanes 1 and 2).

To test the requirement for the other two endoribonucleases, *rnp^ts* and *rne^ts* strains [26,27] were grown to an $OD_{600}$ of 0.3 at a permissive temperature (30°C), together with their respective WT strains. The cultures were then incubated at the non-permissive temperature (44°C) for an additional 25 min before harvesting the cells for analysis. By this time, the culture $OD_{600}$ reached about 0.6 in most cases. In *rnp^ts*, no difference in the levels of the two 3' ends were

observed in cells harvested at the permissive and non-permissive growth temperatures, indicating non-involvement of RNase P in the generation of either of the *galET* species (Fig 3A, lanes 3–6). In contrast, in *rne*^ts at the non-permissive temperature, the level of the 3' end of *galET-long* decreased to less than 10% of the level observed in WT cells (Fig 3A, lanes 8 and 10). In comparison, the difference in the level of the 3' ends of *galET-short* was negligible. These results indicate that RNase E plays a major role in the generation of *galET-long* specifically.

## Rho-dependent transcription termination generates *galET-short* 3' ends

To test the involvement of Rho in the generation of *galET-short*, we assayed transcription by adding pGal in a cell-free system prepared from the *E. coli* strain BL21 (λDE3) [28]. The cell-free system contains all necessary factors for transcription and translation but not template DNA. Reactions were run for 30 min at 30˚C after adding the template DNA, pGal. Results of 3' RACE after adding pGal produced no noticeable 3' ends (Fig 4A, lane 3). However, in the presence of galactose (to a final concentration of 0.5%) produced noticeable but faint 3' ends (Fig 4A, lane 4), indicating that the bands are from transcription of the *gal* operon. Interestingly, the band at 2,184 increased 5 times in the presence of plasmid pRho, where *rho* is under the T7 promoter control (Fig 4A, lane 5). These results show that the 2,184-band is the product of RDT.

To confirm if the band at 2,184 is the result of RDT, we added increasing concentration of the Rho inhibitor Bicyclomycin (BCM) to the cell-free reaction in the presence of pGal, 0.5% galactose and pRho. The band at 2,184 decreased with increasing concentrations of BCM from 0.5 to 10 μg/ml (S2 Fig, lanes 2–6). The BCM at 10 μg/ml inhibited formation of the band at 2,184 by ~90% of the level seen without the drug (S2 Fig, lane 1). These results indicate that the band at 2184 is the result of RDT.

Furthermore, the cytosine at 2,184 is nine nucleotides downstream from a Rho-utilization site (*rut*), known as the "C-rich region", present downstream of the *galT* gene spanning coordinates 2,075 to 2,175 (Fig 4C) [24]. We found a perfect consensus sequence for the RNA polymerase pause site [29,30], in which the cytosine at 2,184 is located at -1 position of the consensus sequence. These results indicate that Rho terminates *gal* transcription paused at 2,184 (Fig 4A).

Next, we tested the effect of Spot 42 RNA in RDT. We prepared Spot 42 RNA using an *in vitro* transcription reaction (See Materials and Methods). We added increasing amounts of Spot 42 RNA to the cell-free reaction together with pGal and 0.5% galactose. Results showed that the band at 2,184 increased with increasing concentration of Spot 42 RNA. The band increased 5, 9, and 12 times when Spot 42 was 1.5, 15, and 150 nM, respectively (Fig 4B). These results do show that Spot 42 directly controls RDT in *gal*.

We observed a cluster of 3' ends at 2,125–2,129 that also increased with increasing Spot 42 concentration. Interestingly, these 3' ends are formed exactly where the 3' ends of *galET-short* are found (Fig 4B). These results indicate that RDT generates *galET-short* 3' ends. These 3' ends are not formed without the added Spot 42 (Fig 4A, lane 5). It appears that the *galET-short* 3' ends are produced by Spot 42-mediated processing of the RDT product at 2,184 by unknown exoribonucleases in the cell-free system.

## RNase III-mediated transcript cleavage results in generation of *galET-short* 3' ends

To identify exoribonucleases that process the 3' ends of *galET-long* and -*short* to their observed locations, we performed 3'RACE on RNA prepared from MG1655Δ*rnb* (RNase II deleted),

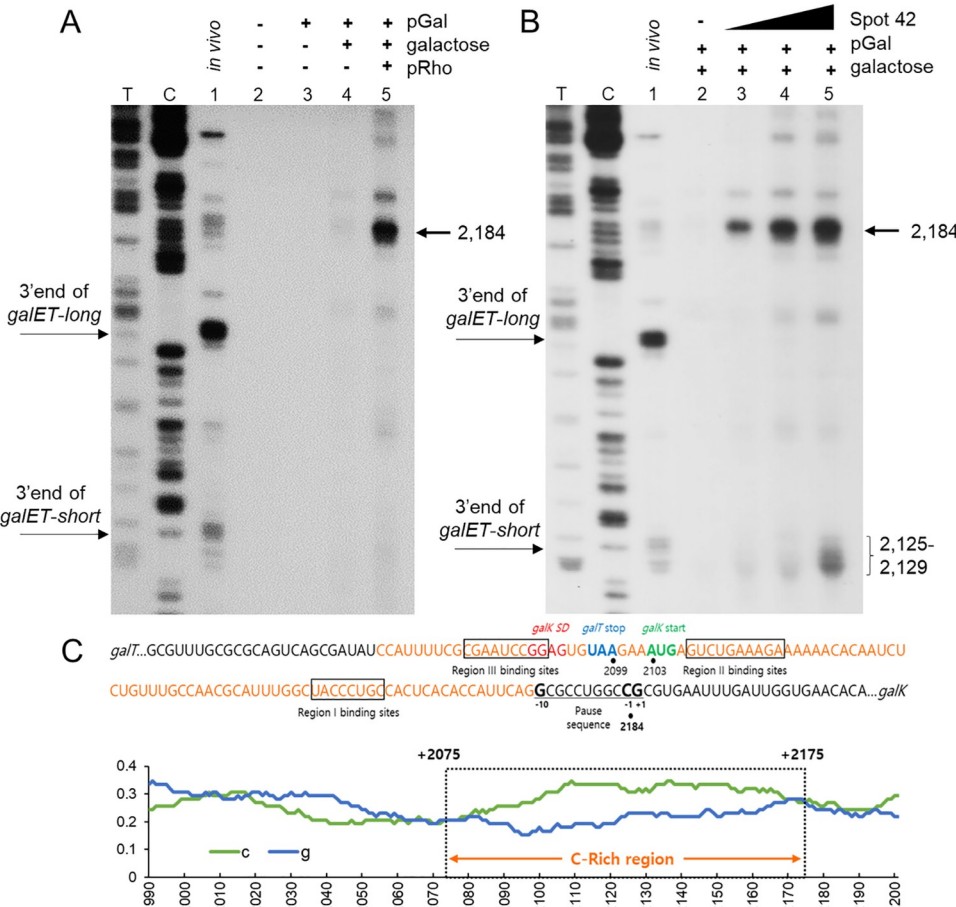

**Fig 4. Rho-dependent transcription termination *in vitro* assayed in a cell-free system.** A) 3'RACE assay of the *galET* mRNA 3' ends from MG1655 *in vivo* (lane 1) and from the cell-free system with no DNA template (lane 2), with DNA template pGal (lane 3), with pGal and 0.5% galactose (lane 4), and with pGal, pRho and 0.5% galactose (lane 5). T, C: DNA sequencing ladders. B) 3'RACE assay of the *galET* mRNA 3' ends from the cell-free system as in (A) except for the absence of pRho plasmid and presence of Spot 42 RNA. Lane 1 is same as in lane 1 of (A); Lane 2 is same as in lane 4 of (A); Lanes 3–5: the same reaction as in lane 2 but with 1.5 nM (lane 3), 15 nM (lane 4) and 150 nM Spot 42 RNA (lane 5). T, C: DNA sequencing ladders. (C) The nucleotide sequence of a Rho-utilization site *rut*, known as the 'C-rich region', which lies downstream of the *galT* gene and spans *gal* coordinates 2,075 to 2,175 (shown in orange). The binding sites in *gal* mRNA for Regions I—III of Spot 42 are boxed. The +1, -1 and -10 of the consensus sequence for the RNA polymerase pause site are presented in bold. Below is shown the cytosine and guanine contents of the region in the green and the blue lines, respectively. The C-rich region (boxed with dotted line) is shown over the abscissa with *gal* coordinates. A sliding window of 70 nt was used to plot the GC content. The data presented in these blots are representative of three independent experiments.

MG1655Δ*rnr* (RNase R deleted) and W3110Δ*pnp* (PNPase deleted). The results show that generation of neither 3' end was compromised in the exoribonuclease deficient strains tested (Fig 5A). This prompted us to test endoribonucleases, specifically the RNA double strand-specific endonuclease RNase III.

We assayed transcription using the cell-free system in the presence of plasmid pRNIII, where the RNase III gene (*rnc*) is under the T7 promoter control. In the presence of RNase III, the 3' ends of *galET-short* at 2,125–2,129 increased 2 times and the 2,184 band decreased by 50 (±5) % (Fig 5B, lane 4). These results further support that the generation of the 3' ends of *galET-short* at 2,125–2,129 require Spot 42 and are processed from the RDT product at 2,184, and unexpectedly, reveal the involvement of an endoribonuclease in this processing.

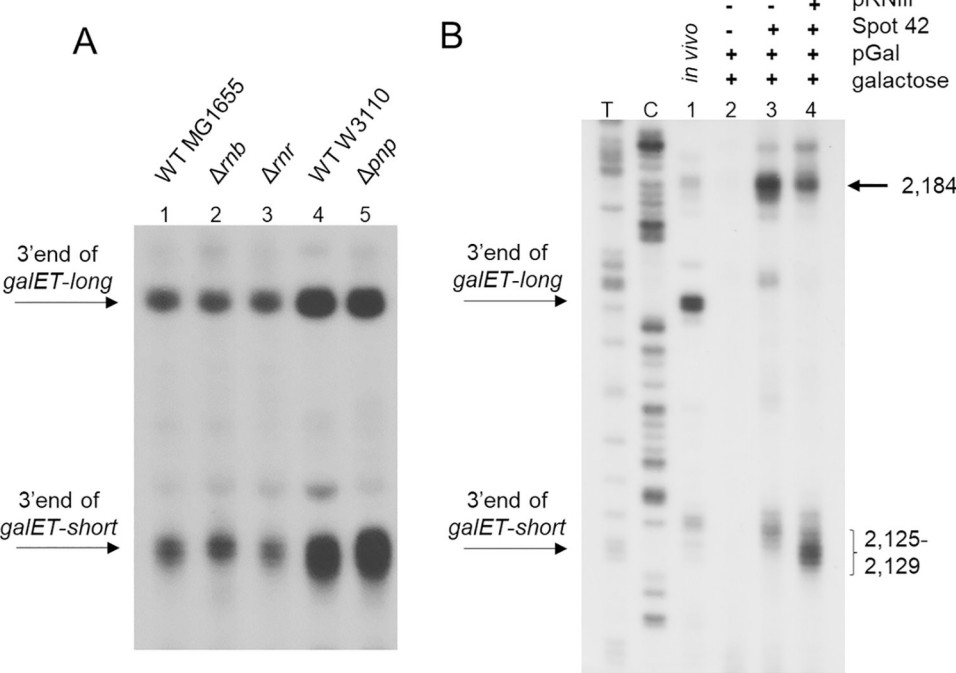

**Fig 5. Generation of *galET-short* 3' ends by RNase III-mediated transcript cleavage.** A) 3'RACE assay of *galET* 3'ends in *E. coli* mutants that are deficient in exoribonucleases. Strains used are MG1655Δ*rnb* (where the gene for RNase II is deleted from the chromosome) (lane 2), MG1655Δ*rnr* (where the gene for RNase R is deleted from the chromosome) (lane 3), W3110Δ*pnp* (where the gene for PNPase is deleted from the chromosome) (lane 5). The results of the WT for strains Δ*rnb* and Δ*rnr* are shown in lane 1, and for Δ*pnp* strain is shown in lane 4. B) 3'RACE assay of the *galET* mRNA from the cell-free system in presence of pRNIII plasmid, where the gene for RNase III is under control of the T7 promoter. The data presented in these blots are representative of three independent experiments.

## Spot 42 tunes production of enzymes for competing pathways of galactose anabolism and catabolism to optimize growth

Spot 42 is known to downregulate *galK* transcriptionally [23] and translationally [20]. Spot 42 is also known to expedite degradation of one of the *gal* mRNA species, *galKM*, which harbors the last two genes of the operon [24]. Here we also found and confirmed that Spot 42 binding leads to the generation of the *galET* mRNA with the two promoter-proximal genes. The expectation from these results is that in the presence of Spot 42 the production of Gal proteins from the promoter-distal genes, *galK* and *galM*, would decrease, while production from promoter-proximal genes, *galE* and *galT*, would increase.

To test this prediction, we measured Gal proteins from MG1655 and MG1655Δ*spf* using western blots. Cells were grown in M9 medium with 0.5% galactose (M9-galactose). The results showed that at OD$_{600}$ of 0.6, MG1655 produced about 83 (±16) % and 68 (±25) % *more* of GalE and GalT proteins, respectively, but 17 (±5) % and 12 (±5) % *less* of GalK and GalM, respectively, than what were produced from MG1655Δ*spf* (Fig 6A). These results indicate that as expected, Spot 42 increases the production of GalE and GalT, and decreases production of GalK and GalM, although the decrease was relatively small. GalE is an epimerase involved in the biosynthesis of the UDP-glucose/galactose required for producing LPS and other polysaccharides, and GalK is a kinase involved in galactose catabolism [31,32]. These results suggest that Spot 42 regulates *gal* expression facilitating the anabolic role of the pathway.

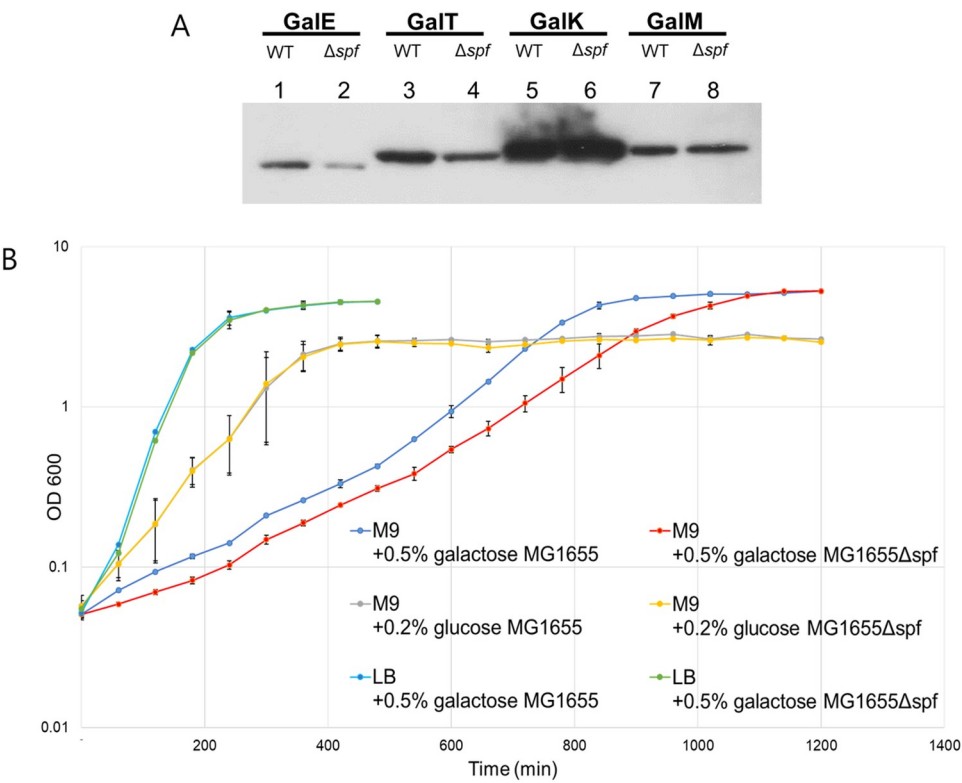

**Fig 6. Spot 42 regulates the tuning of *gal* protein levels towards an anabolic pathway.** A) Western blot analysis of the protein products of *gal* operon (GalE, GalT, GalK and GalM) from WT and *Δspf* cells. B) Physiological effect of Spot 42 deletion. This was measured by comparing growth rate of MG1655 and MG1655*Δspf* cells in various media: LB with 0.5% galactose (light blue and green), M9 with 0.2% glucose (grey and yellow) and M9 with 0.5% galactose (blue and red). The growth of MG1655*Δspf* cells was identical to that of MG1655 in LB-galactose and M9-glucose media. In M9-galactose medium, MG1655*Δspf* cells (red) showed slower growth than that of MG1655 cells (blue). Error bars represents the mean fold-change ± standard deviation of three replicates from three independent experiments (n = 3).

Based on the above results, we reasoned that MG1655*Δspf* cells would grow slower than MG1655 cells in a medium, like M9-galactose, where both anabolism and catabolism of galactose are required. To test this, we measured the growth of MG1655 and MG1655 *Δspf* cells in various media. In LB-galactose or M9-glucose media, where galactose is not required or relevant, respectively, there was no growth difference between the two bacteria (Fig 6B). However, in M9-galactose, MG1655*Δspf* grew slower than MG1655. In exponential growth phase (OD$_{600}$ between 0.05–0.4), the doubling time of MG1655*Δspf* was 23% longer than that of MG1655 (S3A Fig and Table 1).

By the same token, we reasoned that *gal* mutants defective in the generation of *galET* mRNA would grow slower in M9-galactose medium. To test this, we measured the growth of MG1655*Δgal* cells encoding *galMMI-1* (defective in generating *galET-long*), or *galMMII-1* (defective in generating *galET-short*) in M9-galactose medium. In exponential growth phase, the doubling times of cells encoding *galMMI-1* or *galMMII-1* (S3B Fig) were 22 and 26% longer than those encoding the WT *gal* sequence, respectively (Table 1). The growth defect was restored when the *gal* mutants were complemented with Spot 42 mutants that would restore base pairing of Spot 42 with *gal* mRNA (Fig 2F) (S3C Fig). The doubling time of *galMMI-1* and *galMMII-1* in the presence of the Spot 42 mutants with complementary changes became 21 and 22% shorter, respectively, than when Spot 42 was WT (Table 1). These results reveal

**Table 1.  Doubling time of various *gal* and *spf* mutants in M9-galactose medium.**

| Strain / plasmid | Doubling time (min) in exponential phase (OD$_{600}$ 0.05–0.4) |
|---|---|
| MG1655 | 71±3 |
| MG1655 Δ*spf* | 88±5 |
| MG1655 Δ*gal* /pgal | 65±5 |
| MG1655 Δ*gal* /pgalMMI-1 | 79±3 |
| MG1655 Δ*gal* /pgalMMII-1 | 82±6 |
| MG1655 Δ*gal* Δ*spf* /pgal /pSpot42 | 108±11 |
| MG1655 Δ*gal* Δ*spf* /pgalMMI-1 /pSpot42 | 136±8 |
| MG1655 Δ*gal* Δ*spf* /pgalMMI-1 /pSpot42MMI-1 | 107±2 |
| MG1655 Δ*gal* Δ*spf* /pgalMMII-1 /pSpot42 | 144±26 |
| MG1655 Δ*gal* Δ*spf* /pgalMMII-1 /pSpot42MMII-1 | 112±5 |

the physiological relevance of differential expression of the *gal* operon genes by Spot 42, which is to optimize growth by directing *gal* expression towards anabolic pathway.

## Discussion

### Transcript cleavage and transcription termination are both involved in the generation of *galET* mRNA

In this study, we explored mechanisms by which a noncoding sRNA, Spot 42, regulates expression of the *gal* operon of *E. coli* at post transcription-initiation level. Our results indicate that in the presence of Spot 42, two *galET* species are generated. These mRNAs have the same 5' end as the full-length *galETKM* mRNA [33], but have two different 3' ends, herein referred to as *galET-short* and *galET-long*. These RNAs are generated by base pairing of two different regions of Spot 42 (regions I and II) with their complementary sequences near the beginning of *galK* ORF. Region II base pairing leads to the formation of *galET-short* and requires the transcription termination factor Rho, and region I base pairing leads to the formation of *galET-long* and requires the endoribonuclease RNase E. The involvement of Rho in *galET* generation was known and that it is also generated by RNase E is reported here. Below, we discuss how base pairing in one region brings about transcription termination and, in another region, transcript cleavage.

Transcription is coupled to translation in *E. coli* [34–36]. When coupled, RNA polymerase is physically connected to the leading ribosome that translates the nascent transcript [37–39]. This coupling is a stochastic event, such that RNA polymerase often transcribes an ORF without a linked ribosome, generating regions of ribosome-free mRNA [40,41].

Based on the experimental results of Moller et al [20], which show that Spot 42 binding inhibits translation initiation of *galK*, we propose that when transcription is stochastically uncoupled from translation, Spot 42 initiates base-pairing with *gal* mRNA and particularly the region II base-pairing immediately downstream of the *galK* start codon blocks the initiation of *galK* translation (Fig 7A) [42]. The resultant generation of ribosome-free RNA, which includes an Rho utilization site (*rut*), known as a 'C-rich region' [24], allows Rho binding to the *rut* site and subsequent termination of transcription that is paused at 2,184 (Fig 7B) [1,15,17,43,44]. RNase III is involved either alone or in conjunction with an unknown exoribonuclease in the processing of Rho-terminated transcript with 3' end at 2,184 to *galET-short* products with 3'ends at 2,125–2,129 (Fig 7C).

Unlike Rho loading, which depends on uncoupling of transcription from translation, cleavage by RNase E does not have to be so restricted. It is likely that both nascent and non-nascent

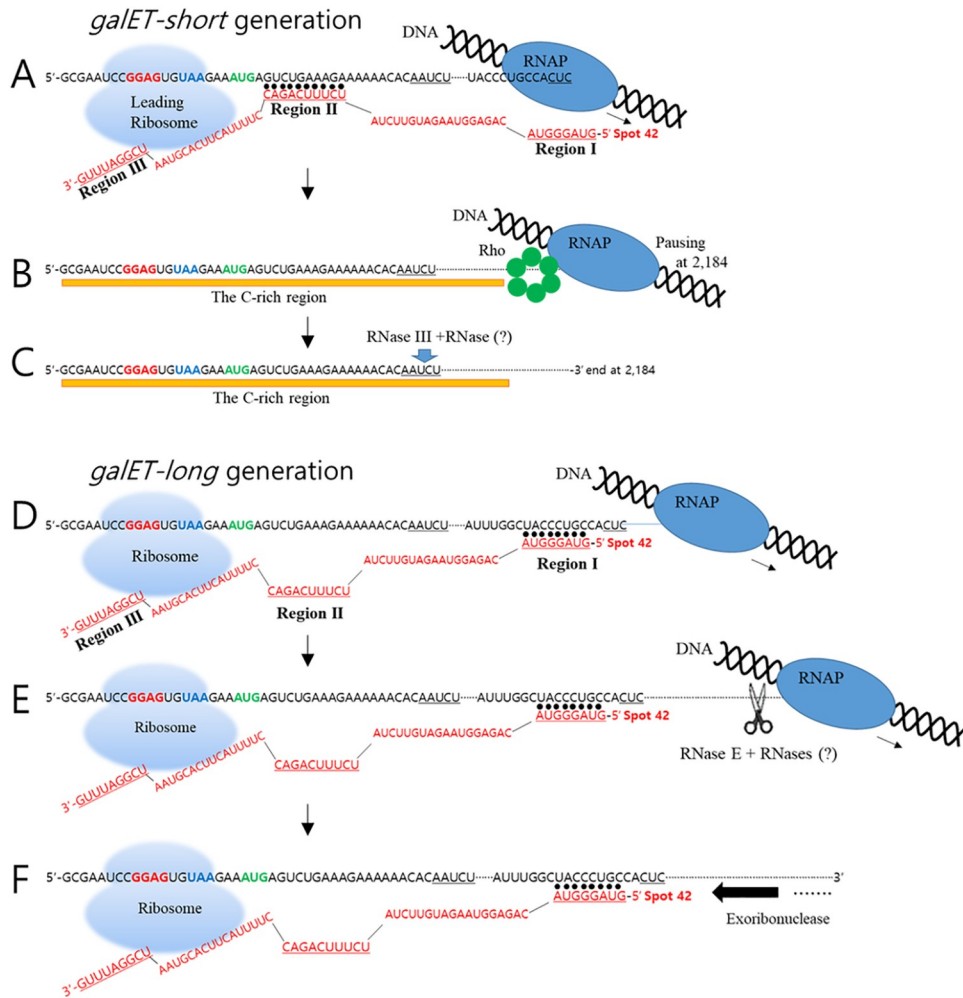

**Fig 7. Model of Spot 42 sRNA-mediated generation of *galET* RNA species by Rho and RNase E.** A) Generation of *galET-short* by Rho: The model shows the (leading) ribosome at the stop codon of *galT* on a nascent chain of *gal* mRNA at the *galT-galK* intercistronic region uncoupled from a transcribing RNA polymerase. In this situation, the mRNA sequences that pair with Spot 42 region II are not covered by the ribosome. B) The Region II binding is assumed to maintain downstream mRNA ribosome-free. This provides the platform for Rho loading and generation of *galET-short* by transcription termination by Rho at 2184. The 3'end of *galET-short* is underlined. C) The subsequent processing of the Rho-terminated transcript by RNase III cleavage is shown at the 3'end of *galET-short* (downward green arrow). D) Production of *galET-long* by RNase E: This process requires hybridization with Spot 42 Region I. We assume that translation termination at *galT* does not always lead to translation initiation of *galK* (due to stochastic uncoupling of *galK* translation from *galT* translation), which would keep *galK* RNA ribosome-free and promote hybridization with Region I. (E, F) The hybridization is assumed to ensure that the downstream RNA remains ribosome-free in the nascent transcript (E) or in non-nascent transcripts (F) to allow RNase E cleavage. The hybrid Region I could also block 3'-end processing by 3'→ 5' exonucleases after RNase E cleavage. Note that Spot 42 Region III cannot base pair with the corresponding *gal* sequences in this situation due to the presence of the ribosome at *galT* stop codon, which might explain why mutations in that region are inconsequential to *galET* mRNA production. "RNase (?)" represents currently unidentified RNases(s) that may be involved in the further processing events of the *galET* transcripts.

transcripts could serve as substrates for Spot 42-mediated *galET-long* production. The probability of cleavage by RNase E would thus depend on the probability of uncoupling of translation from transcription in the case of nascent transcripts, and the uncoupling of *galK* translation from *galT* translation in non-nascent transcripts (Fig 7D–7F). Spot 42 binding to transcripts is expected to increase the probability of downstream RNA staying ribosome-free,

thus facilitating RNase E action (Fig 7E). Spot 42 could also recruit RNase E by allowing RNase E to bind to the 5' mono-phosphorylated end of Spot 42 [45]; however, this is not supported by our data (S4 Fig). It appears that RNase E is accessing ribosome-free mRNA by the direct entry pathway [45].

In summary, our results demonstrate that like transcription termination, transcript cleavage by RNase E is another mechanism to generate *galET* mRNA species. We believe that this is the first example of a bacterial sRNA generating essentially a single mRNA product using at least two different mechanisms. Simultaneous regulation by both RNase cleavage and RDT, however, has been described for riboswitches [46–48].

Our data also suggest that the two events that generate *galET* happen independently, i.e., the frequency with which one event occurs does not affect the frequency with which the other occurs (Fig 2B and 2C). Although these events could influence each other by depleting substrates, we note that a significant fraction of the substrates (*galETKM* and *galETK*) remain intact (Fig 1B), suggesting that the substrate availability may not be a limiting factor. We believe both events are probabilistic and occur on different transcripts with comparable efficiencies. These results can also be understood if Spot 42 exists in two forms that are not in equilibrium. In that case, loss of one will not affect the outcome from the other.

## Processing of the 3'-OH ends generated by transcription termination or transcript cleavage

The Spot 42 region I mutant Spot42DMI-1 (which has a single nucleotide deletion at the 5' end) generates *galET-long* mRNA that is two nucleotides shorter at the 3' end (Fig 2B, lane 2). One possible explanation of these results is that the base-paired double stranded RNA between region I and *gal* mRNA blocks 3'→ 5' exoribonuclease digestion, because it is well known that a hairpin (stem-loop) structure at the 3' end of *E. coli* mRNA can protect the free 3'-OH end from such digestion [49–52].

The 3' ends of RDT-generated transcript are known to be immediately subjected to 3'→ 5' exoribonuclease digestion [4,50]. For the first time, however, we report in this study that RNase III, which is a double-stranded RNA specific endoribonuclease, can process the 3' end of a transcript that is released from RDT (Fig 5). It is not clear, however, whether the RNase III cleavage sites are on the double stranded RNA region formed between Spot 42 and *gal* RNA or that formed intramolecularly in *gal* RNA sequences.

## sRNA binding at intercistronic region and at 5' UTR may have different outcomes

sRNA binding to the 5'untranslated region (5'UTR) generally causes inhibition of translation initiation. This results in generation of ribosome-free region downstream from the sRNA binding region, which could allow either transcription termination by Rho [17,24,53,54] or transcript cleavage by RNase E [7–12]. Usually, a new mRNA 3' end generated by either of these events becomes immediately subjected to 3'→5' exoribonuclease digestion, which results in the degradation of target mRNA.

However, if sRNA binds to intercistronic regions of a multi-cistronic mRNA, this sRNA could inhibit the initiation of translation of the downstream gene, making recruitment of Rho or RNase E possible. In these cases, 3'→5' exoribonuclease digestion (beginning at the newly generated 3' end) is likely to stop at the sRNA-mRNA hybrid and not likely to reach the 5' end of the target mRNA and degrade the message to completion. Thus, sRNA binding at the intercistronic region in multi-cistronic mRNA could protect upstream cistrons, resulting in the accumulation of relatively stable new mRNA species. We observed an example of this in our

study: Spot 42 binding at the intercistronic region of *galT-galK* led to the generation of a new mRNA species, *galET*. Considering the prevalence of potential sRNA targets at cistron junctions [55], the generation of new mRNA species by the mechanisms described here may be a general function of sRNA in polycistronic gene regulation.

## Relationship of cAMP, Spot 42 and polarity in *gal* operon

Polarity in *gal* operon is conspicuous in glucose medium in WT cells (where cAMP level is low) or in cAMP mutant cells [2,56–58]. These two observations indicate that cAMP is an inhibitor of polarity. Spot 42 synthesis is repressed by cAMP and is induced in glucose medium or in cAMP mutant cells. In other words, polarity in *gal* operon seen in the absence of cAMP can be attributed to the induction of Spot 42.

## Materials and methods

### Bacterial strains and primers

*E. coli s*trains used were GW10 (*rng+*, *rne+*; WT for RNase G and RNase E), GW11 (*rng::cat*; RNase G down mutant), GW20 (*ams1^{ts}*; RNase E temperature-sensitive mutant), NHY312 (*rnpA+*; WT for RNase P), and NHY322 (*rnpA49*; RNase P temperature-sensitive mutant). Chromosomal deletion strains of *E. coli* MG1655 were generated using phage Lambda Red-mediated recombineering [59]. Bicyclomycin (BCM) was a generous gift from M. Gottesman (Columbia University, USA). Primers used in this study are listed in S1 Table. Bacterial strains used in this study are listed in S2 Table.

### Bacterial growth conditions

All cells were grown at 37˚C in Lysogeny Broth (10 g tryptone, 5 g yeast extract, and 10 g NaCl per liter of water) supplemented with 0.5% (w/v) galactose and chloramphenicol (15 μg/ml) or ampicillin (100 μg/ml) to an $OD_{600}$ of 0.6 before RNA isolation. To determine generation time during exponential growth, cultures in M9-galactose grown to $OD_{600}$ of 1.0 were diluted to a fresh medium to $OD_{600}$ of 0.01, and growth was monitored by taking $OD_{600}$ periodically up to ~0.4.

### RNA preparation

Total RNA was purified from clarified cell lysates using the Direct-zol RNA MiniPrep kit (Zymo Research, USA). To generate cell lysates, $2 \times 10^8$ cells were resuspended in 50 μl of Protoplasting buffer (15 mM Tris-HCl [pH 8.0], 0.45 M sucrose and 8 mM EDTA). Five μl of lysozyme (50 mg/ml) was added and the sample was incubated for 5 min at 25˚C. A phenolic detergent (1 ml TRI Reagent; Molecular Research Center, USA) was added and vortexed for 10 sec before incubating for 5 min at 25˚C. The RNA was isolated according to the manufacturer's recommendations and dissolved in 30 μl of RNA storage buffer (Thermo Fisher Scientific, USA). RNA concentrations were determined by measuring the absorbance at 260 nm using a NanoDrop spectrophotometer (Thermo Fisher Scientific, USA).

### Northern blot analysis

Total RNA (10 μg in 10 μg/ml ethidium bromide) was resolved by 1.2% (w/v) formaldehyde-agarose gel electrophoresis at 8 V/cm for 2 h. RNA integrity was assessed under a UV light and was transferred overnight to a positively charged nylon membrane (Thermo Fisher Scientific, USA) using a downward transfer system (Whatman TurboBlotter, USA). The RNA was fixed to the nylon membrane by baking at 80˚C for 1 h. The northern blot probes were prepared as

follows. First, a 500-bp DNA fragment from the *galE* region (from +27 to +527) was prepared by PCR, followed by labeling with $^{32}$P. The template DNA (0.15 pmol) was mixed with random hexamers (4 μl of 1 mM) (Takara, Japan) in a total volume of 28 μl and then heated at 95˚C for 3 min. The reaction was then rapidly cooled for 5 min on ice and a mixture containing 5 μl of 10× Klenow buffer, 5 μl deoxynucleoside triphosphate (dNTP) mix (0.2 mM dATP, dGTP, and dTTP), 10 μl α$^{32}$P-dCTP, and 2 μl Klenow fragment (2 U/μl) (Takara, Japan) was added. The reaction was incubated at 37˚C for 1 h, followed by inactivation of the Klenow fragment at 65˚C for 5 min. The product was purified by passage through a G-50 column. The blot was pre-hybridized in 7 ml ULTRAhyb Ultrasensitive Hybridization buffer (Invitrogen, USA) at 65˚C for 30 min. The DNA probe was denatured at 95˚C for 5 min and 5 μl of the probe was added to the hybridization buffer. The hybridization was performed at 42˚C overnight, and the blot was then washed twice in low-stringency wash buffer (2× SSC, 0.1% SDS) for 5 min each at room temperature and then twice in high-stringency wash buffer (0.2×SSC, 0.1% SDS) for 15 min each at 42˚C. Finally, the radioactive bands were visualized after exposure to X-ray film. The films were scanned and RNA bands were quantified using ImageJ software (NIH, USA).

## Rapid amplification of cDNA ends (RACE)

Contaminating DNA in the RNA preparation was removed using Turbo DNase I (Thermo Fisher Scientific, USA). For the amplification of the 3' RNA end (3' RACE) or the 5' RNA end (5' RACE), we performed an RNA ligation reaction in a 25 μl reaction volume containing 1 μl of a 100 nM synthetic RNA oligo (27 mer for 3'RACE (S1 Table) and *E. coli* 5S rRNA for 5' RACE), 2.5 μg or 5 μg total RNA for 3'RACE or 5'RACE, respectively, 2.5 μl 10x reaction buffer (Thermo Fisher Scientific-USA), 10 U T4 RNA ligase (Thermo Fisher Scientific-USA), and 10 U of rRNasin Ribonuclease Inhibitor (Promega, USA). The ligation reaction was incubated at 37˚C for 3 h, and the resulting RNA was purified using a G-50 column (GE Healthcare, USA). Four micrograms of the ligated RNA was reverse transcribed at 37˚C for 2 h in a 20 μl reaction mixture containing 4 U of Omniscript reverse transcriptase (Qiagen, Germany), 0.5 mM of each dNTP, 10 μM random hexamers and 10 U of rRNasin. For 3' RACE, the specific 3RP primer (S1 Table), which hybridizes to the ligated RNA, was used. Either 2.5 or 10 μl was used as the template for PCR amplification of the target RNA for 3'RACE or 5'RACE, respectively, using a target RNA specific primer set and HotStarTaq DNA polymerase (Qiagen, Germany) in a 50 μl reaction volume. The amplified cDNA was purified and used as a template for the primer extension reaction, which was performed in a 20 μl volume with a $^{32}$P-labeled specific primer and one unit of Taq polymerase (Qiagen, Germany) as previously described [2]. The reaction products were resolved on an 8% (w/v) polyacrylamide-urea sequencing gel, and the radioactive bands were visualized using X-ray films.

## Reverse Transcription-quantitative PCR (RT-qPCR)

Reverse transcription was performed after removal of genomic or plasmid DNA from the reaction mixtures using Turbo DNase I according to the manufacturer's recommendations (Thermo Fisher Scientific, USA). One microgram of total RNA was incubated at 37˚C for 2 h in a 20-μl reaction volume containing 4 U of Omniscript reverse transcriptase, 0.5 mM of each dNTP, 8 μM random hexamer primer, and 10 U of rRNasin. Using a CFX96 PCR instrument (Bio-Rad, USA), RT-qPCR was performed in a 10 μl reaction volume containing 5 μl of iQ SYBR Green Supermix (Bio-Rad, USA), 3 μl of RNase-free water, 0.5 μl each of 10 μM forward and reverse primers, and 1 μl of the cDNA template. The following conditions were applied: an initial denaturation step at 95˚C for 3 min and 40 cycles of 10 s of denaturation at 95˚C, 20

s of hybridization at 60˚C, and 10 s of elongation at 72˚C. The results from each sample were normalized using rrsB, which encodes 16S rRNA.

## Mutagenesis and quantification of Spot 42

Spot 42 mutations were generated in the pSpot42 plasmid [23] using the mega-primer method [60]. All mutations were confirmed by DNA sequencing. The 5'RACE method was used to quantify Spot 42 production.

## *In vitro* synthesis and purification of Spot42 RNA

*In vitro* synthesis of Spot42 RNA was performed with T7 RNA Polymerase (New England Bio-Labs, USA) according to the manufacturer's instructions. The DNA template containing the T7 promotor was generated by PCR using the T7-spot42-for and T7-spot42-rev primers using pSpot42 plasmid [20] as the template. Briefly, DNA template (1μg) was incubated at 37˚C for 1 h in reaction buffer (40 mM Tris-HCl, pH 7.9; 6 mM MgCl2; 2 mM spermidine; 1 mM dithiothreitol [DTT]) containing 2 U RNA polymerase, 1 U RNase inhibitor and 10 μl rNTP mix (final concentration, 2 mM each NTP) in a 50 μl reaction. Post-incubation at 37˚C for 1 h, contaminating DNA in the reaction was removed by adding Turbo DNase I and incubation at 37˚C for 15 min. The reaction was stopped by adding 5 μl of DNase I Inactivation reagent (Thermo Fisher Scientific, USA). After vortexing and centrifugation at 2,650 g for 5 min, the aqueous phase (30 μl) was passed through a G-50 column. The purified RNA was quantified and stored at -70˚C.

## Assay for Rho dependent transcription termination using the cell-free expression system (*in vitro* transcription translation reaction)

We used the pGal plasmid as DNA template for *in vitro* transcription-translation reaction. The reaction was performed using *E. coli* extract (S30) containing *E. coli* RNA polymerase holoenzyme/T7 RNA polymerase (Bioneer, Korea) for transcription and all essential components for translation [28], according to the manufacturer's instructions. In short, DNA template (10 nM) and pRho plasmid (where rho is under the T7 promotor/terminator and RBS [Ribosomal Binding Site] control) was incubated at 30˚C for 30 min containing the *E. coli* extract in reaction buffer (Bioneer, Korea) (that includes including amino acids, rNTPs, tRNAs, an ATP generating system, IPTG and appropriate salts) with 0.5% (w/v) galactose in a 60 μl reaction. Post-incubation at 30˚C for 30 min, the reaction was stopped by adding equal volume of phenol: chloroform: isoamyl alcohol 25:24:1 (Merck, USA). After vortexing and centrifugation at 2,650 g for 10 min, the aqueous phase (30 μl) was passed through a G-50 column. The purified nucleic acids were used for 3'RACE reactions.

## Western blot analysis

For the western blot analysis of the Gal proteins, we inserted in frame a 'His-tag' (a string of histidine residues) at the end of the open reading frame of each *gal* gene of the *gal* operon DNA in the plasmid pGal, generating plasmids; pGalE-histag, pGalT-histag, pGalK-histag, and pGalM-histag. MG1655 or MG1655Δ*spf* strain harboring each one of the plasmids was grown in M9-galactose to OD600 of 0.6. Proteins in crude cell lysate from each culture (5 μg) were separated on a SDS-acrylamide slab gel electrophoresis and blotted on a nitrocellulose membrane using a protein blotter (Bio-Rad, USA). The blots were probed with the His-tag antibody (Merck, Germany) according to manufacturer's instructions.

## Supporting information

**S1 Fig. Correspondence of intensities of *galET* bands between northern blots and 3'RACE analysis results.** In the absence of Spot 42, particularly the *galET* band in (*A*) disappears from the northern blot (lane marked Δ*spf*), as was seen in Fig 1B. Similarly, in Δ*spf* cells, generation of the 3' ends of *galET-short* and *-long* is significantly (~ 95%) inhibited. When Spot 42 was overproduced (*spf*\*), the *galET* production increased 150% of WT (*A*), and the production of the 3' ends of *galET-short* and *-long* also increased correspondingly (*B*).
(TIF)

**S2 Fig. Rho-dependent transcription termination *in vitro* assayed in a cell-free system.** 3'RACE assay of *galET* mRNA 3' ends from the cell-free system with pGal, pRho and 0.5% galactose, and either with no BCM (lane 1) or increasing concentrations of BCM (0.1, 0.5, 1, 5 and 10 μg/ml) (lanes 2–6).
(TIF)

**S3 Fig. Growth rate of various cells in the exponential phase in the M9-galactose medium.** (*A*) MG1655 (blue) and MG1655 Δ*spf* (red). (*B*) MG1655Δ*gal* /pGal (blue), MG1655Δ*gal* /p*galMMI-1* (green) and MG1655Δ*gal* /p*galMMII*-1(red). (*C*) MG1655Δ*gal*Δ*spf* /pGal/ pSpot42 (blue), MG1655Δ*gal* Δ*spf* /p*galMMI-1*/pSpot42 (brown), MG1655Δ*gal*Δ*spf* /p*galMMI-1*/pSpot42MMI-1 (gray), MG1655Δ*gal*Δ*spf*/p*galMMII-1*/pSpot42 (green) and MG1655Δ*gal*Δ*spf* /p*galMMII-1*/pSpot42MMII-1(red).
(TIF)

**S4 Fig. 3' RACE assay of *galET* 3'ends in MG1655Δ*rppH*, a strain deficient in RNA pyrophosphohydrolase.** RNase E binds to mono-phosphorylated 5' end of sRNA. This ability enables RNase E to be recruited to the site of cleavage on target mRNA (1). To see if RNase E cleavage that generates *galET-long* depends on mono-phosphorylated 5' end of Spot 42, we assayed the 3' ends of *galET* in MG1655Δ*rppH* strain where the gene for the RNA pyrophosphohydrolase is deleted from the chromosome. Without the RNA pyrophosphohydrolase, most of the 5' end of RNA remains in di-phosphorylated state [33]. Results showed no difference in generation of the 3' end of *galET-long* and *-short* between WT (lane 1) and MG1655Δ*rppH* (lane 2) strains. These results demonstrate that the mono-phosphorylated 5' end of Spot 42 is not a requirement for the generation of *galET-long*.
(TIF)

**S1 Table. Primers used in this study.**
(DOCX)

**S2 Table. Bacterial strains used in this study.**
(DOCX)

## Acknowledgments

Authors are grateful to Prof. Gerhart Wagner (Uppsala U, Sweden) and Dr. Nadim Majdalani (NIH, USA) for their thoughtful comments. Authors thank Jeongok Park for cell growth measurements.

## Author Contributions

**Conceptualization:** Heon M. Lim.

**Funding acquisition:** Dhruba K. Chattoraj, Heon M. Lim.

**Investigation:** Heung Jin Jeon, Yonho Lee, Monford Paul Abishek N, Xun Wang, Dhruba K. Chattoraj.

**Methodology:** Heung Jin Jeon, Yonho Lee, Monford Paul Abishek N, Xun Wang.

**Supervision:** Heon M. Lim.

**Validation:** Heung Jin Jeon, Yonho Lee, Monford Paul Abishek N, Xun Wang, Dhruba K. Chattoraj, Heon M. Lim.

**Writing – original draft:** Dhruba K. Chattoraj, Heon M. Lim.

**Writing – review & editing:** Dhruba K. Chattoraj, Heon M. Lim.

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
