## [Decision Letter · Decision Letter 0]

14 Jun 2021

Dear Dr Lim,

Thank you very much for submitting your Research Article entitled 'sRNA‑mediated regulation of gal mRNA in E. coli: Involvement of transcript cleavage by RNase E together with Rho‑dependent transcription termination' to PLOS Genetics.

The manuscript was fully evaluated at the editorial level and by independent peer reviewers. The reviewers appreciated the attention to an important problem, but raised some substantial concerns about the current manuscript. Based on the reviews, we will not be able to accept this version of the manuscript, but we would be willing to review a much-revised version. We cannot, of course, promise publication at that time.

If you decide to revise the manuscript for further consideration at PLOS Genetics, please aim to resubmit within the next 60 days, unless it will take extra time to address the concerns of the reviewers, in which case we would appreciate an expected resubmission date by email to plosgenetics@plos.org.

[LINK]

We are sorry that we cannot be more positive about your manuscript at this stage. Please do not hesitate to contact us if you have any concerns or questions.

Yours sincerely,

Jue D. Wang

Associate Editor

PLOS Genetics

Josep Casadesús

Section Editor: Prokaryotic Genetics

PLOS Genetics

Reviewer's Responses to Questions

**Comments to the Authors:**

Reviewer #1: This is a beautifully written manuscript dealing with the regulation of the E. coli gal operon by Spot 42, an sRNA. The findings are surprising and quite novel. Spot 42 promotes Rho-dependent termination at galK as well as degradation of the galK transcript. Amazingly, these effects are performed by different SpoT 42 sequences, both complementary to galK mRNA.

Critique:

1. There are several typos that need correcting.

2. Eliminating one of the two SpoT activities does not increase the activity of the other, suggesting that the substrate mRNA is in excess. It may suggest that SpoT has two structures that are not in equilibrium. Is anything known about the structure of this sRNA?

3. The overall efficiency of SpoT processing is very low. I wonder how a SpoT could affect cell growth. And whether it plays a significant role in Gal regulation.

4. If would be useful to indicate the extent of processing under the gel lanes (Fig 1).

5. Fig 3C needs significance calculations - what is Fig3D looking at? Also, what does the time course mean – it the values are quite variable.

6. Fig 4 Is the lettering correct? The band at 2184 doesn’t correspond to anything in vivo. What does this figure add to understanding the mechanism of action of SpoT? Unless you can explain why it isn’t seen in vivo.

7. Fig 5 – statistical significance not given.

Reviewer #2: I have reviewed the manuscript by Jeon et al. submitted to Plos Genetics. The manuscript basically reports investigation of mRNA metabolism originating at the gal operon in E. coli. Specifically, the authors deal with post-initiation processing of RNA and transcription termination, and the regulation of these events. More than 50 years ago, a phenomenon was discovered, called Natural Polarity, which is decreased expression of distal cistrons compared to the expression of promoter proximal cistrons in a polycistronic operon. Regulations both at the transcriptional and translational level have been independently suggested to be responsible for natural polarity but the exact mechanism(s) of natural polarity remained unproven for decades. For the first time, by in depth analysis of the gal operon in E. coli the current authors have demonstrated that the phenomenon is complex and occurs at different levels: transcription termination, RNA processing and translation. A forerunner paper analyzing natural polarity was published in 1980s in the his operon of Salmonella although determining the precise mechanisms was not possible at the time because of lack of modern technologies. In the current manuscript, Jeon et al. showed how a small RNA participates in creating natural polarity in all three steps – transcription termination, RNA processing and translation. It is also noteworthy that involvement of small RNA in Rho-mediated transcription termination is novel.

In gal, the natural polarity is condition dependent – only shown in the absence of cAMP and has huge physiological implications.

The experiments are done in meticulous details with clear and convincing results. Importantly, the authors also nicely correlate the implications of the mechanisms with physiological experiments. I strongly recommend that the manuscript should be published as soon as possible. However, I suggest that several minor editorial corrections before publication. They are:

Line 63. Mention the 3’ endpoint if known.

Line 101. Delete the phrase “rather than by new synthesis”

Line 170. It appears that region III of the small RNA does not have a role. The authors may speculate why not.

Lines 241-3. Bicyclomycin does not inhibit Rho activity completely. It is suffice to see A partial effect of the drug is expected. It is not necessary to invoke another factor.

Line 248. Use the phrase ‘in the presence of’ of ‘adding’.

Line 251. Same. ‘In the presence’ of plasmid pRho instead of ‘adding’.

Lines 260 and 267. Same. ‘In the presence’ of pGal, etc.

Line 270. ‘in the galK gene sequence’.

Line 300. …. at ‘the’ galT…

Line 313. Here ‘we’ also found….

Line 317. “Increased”?

Line 327. It is better to say ‘…. Expression facilitating the anabolic role of the pathway’.

Starting line 328. Great experiments showing the physiological relevance of spot 42 in gal operon expression.

Line 352. Extend the sentence ‘E. coli .. at post transcription initiation level.

Line 373. The unknown enzyme could be RNase II as shown by the authors (ref. 4).

Lines 374-377. See comments on lines 241-243.

Line 410. I agree. PNPase acts the same way as RNAse II does.

Line 433. …in ‘polycistronic’ gene regulation.

Discussion. It is bit long, and may be somewhat condensed by removing a few of the ‘remains to be investigated’ topics.

Discussion. I suggest that the authors mention the relationship of cAMP and polarity in gal that I discuss above simply because the relationship has been unraveled in the paper.

Reviewer #3: The manuscript by Jeon et al investigates regulation of the E. coli gal operon by Rho, RNaseE, and a small RNA Spot 42. This work builds on previous observations published in 2014 and 2015. The authors show that binding of Spot 42 to the gal mRNA leads to Rho-dependent termination (that was already shown by the authors) and RNase E cleavage at two closely-spaced sites.

In my opinion, the insights provided by this work fall short of the PLOS Genetics standards.

Binding of Spot 42 to the TIR of galK is expected to inhibit translation, making RNA susceptible to Rho and RNA degradation. In their previous PNAS paper, the authors proposed just that. The new data presented in this work are (i) effects of mutations in mRNA, sRNA, and compensatory mutations on the formation of two mRNA species and (ii) the involvement of RNaseE, but not RNaseP or G, in mRNA processing. With respect to Rho, the authors add a lot of small new pieces, including in vitro coupled assay in which Rho is expressed. However, nothing principally new is shown and their previous conclusions do not change. It is well established how Rho works. On that note, the authors should brush up on Rho literature – papers they cite on the mechanism of action of BCM and Rho itself do not reflect current knowledge in the field, and their discussion does not incorporate recent findings. Similarly, the collection of transcription-translation references is really odd.

By contrast, data on RNAseE are limited to one gel with a ts mutant. If the authors want to make their study impactful, they should investigate this aspect of the regulatory mechanism.

With respect to Spot 42, the authors should inhibit translation of galK by other means (many possibilities) to distinguish different aspects of its contribution to gal control. The authors conclude that Spot 42 “directly” controls termination based on data in Fig. 4 – this is impossible to tell in the assay used, as all RNA synthesis goes up upon the addition of Spot 42 RNA, not only the species authors mention. In this and other assays, a normalization control is essential.

Some aspects of other assays are puzzling. For example, in Fig. 3, the authors argue that a small effect of Rho inhibitor BCM after extended incubation is supportive of their conclusions. Rho terminates 20% genes in E. coli, and BCM leads to a lot of secondary effects; particularly after 1 hour. In this case, error bars would be critical but are absent.

The authors mention the use of strain with a non-functional Rho – they must elaborate as rho is an essential gene in E. coli unless RNase H/UvsW is overexpressed.

Comparing reporter fusions to tRNA-Arg gene +/-rut makes no sense. One should do RT PCR probing regions before and after terminator. And the effects presumably due to termination are tiny! There are tens of well-characterized mutants in Rho and NusG that could be used to probe the role of Rho.

Making physiological arguments based on plasmids that express regulatory factors is not acceptable. MG1655 can be genetically modified as needed.

Most importantly, the authors discuss possible effects of uncoupling of transcription and translation and the roles of small RNAs. There is no doubt that sRNAs have a lot of targets and control expression of many genes. But the effects reported here are small, not necessarily direct, and need to be investigated in more detail.

Reviewer #4: In this work, Jeon and Lee et al. build upon previous analyses of regulation in the well-studied E. coli gal operon. The authors present the interesting finding that both RNase E and Rho, in conjugation with the sRNA, Spot 42, control formation of alternative forms of the galET sub-transcript (long and short forms, respectively). In particular, base-pairing between Spot 42 region I and the gal transcript favors generation of galET-long, and base-pairing between Spot 42 region II and the gal transcript favors generation of the galET-short transcript. galET-short generation is dependent upon Rho factor activity, whereas galET-long generation is dependent on RNase E activity. Finally, through specific base-pairing interactions, Spot 42 is predicted to help direct metabolites to anabolic pathways during growth in minimal medium. Overall, this work should be of interest to the field, but the manuscript requires both major (including one experiment) and minor revisions as outlined below.

The major revisions needed for this work center around three points. The first point is that information about biological replicates and numbers of blots represented by those presented should be included in the manuscript. The second point centers around Spot 42-mediated, Rho-dependent termination in vivo and whether Fig 5 is sufficient to establish this mechanism, including a need for information about statistical tests used. It is unclear if Spot 42 region I also contributes to Rho-dependent termination from the data presented (i.e., whether RDT may still be promoted via Spot 42 region I, but processing of the gal transcript to galET-short fails without region II base-pairing). Finally, the data presented in Fig 6 seem to test a prediction that was already known to be true rather than directly testing the contributions made by regions I and II to Gal protein level tuning to add biological relevance to the differential processing mechanism described in this work.

Major revisions:

1. The authors do not provide clear statements about the number of biological replicates used to obtain data presented in the figures (e.g., describing the number of replicate blots representative of blots shown represent). This information is especially important for instances where differences are modest and, as such, should be included. In the same vein, it is also not clear where the data for Fig 3D were derived. In particular, were the mean and SD determined from the intensities in Fig 3C? It may be more informative to show comparisons at each time point (e.g., as a line graph), using p-value adjustments for multiple comparisons (error bars should also be included on each point in Fig 3C—the comparisons would, thus, be between galET-short values at each time point or galET-long values at each time point only, rather than between the two isoforms at each time point). As shown in the same figure, the decrease of galET-short over time seems unclear (in fact, from 10 to 20 minutes the level seems to be increasing; the level at 30 minutes is similarly higher than at 10 minutes, and the levels at 40 and 50 minutes seem to be similar to that at 10 minutes). The authors should also mention that the concentration of BCM used is sub-inhibitory and thus is unlikely to completely abrogate RDT. Finally, the authors should provide an explanation for this ambiguous galET-short depletion over time in the text.

2. The authors should revise figure captions to clearly define the data shown in the figures (i.e, with the exception of Fig 3, it is unclear that bars or points represent a mean with error bars representing SD [how many replicates] or some other metric of variability). In Fig 5, are error bars shown for pHL2000? In the same figure, statistical tests should be performed and reported for the comparison of pHL2000 and pHL2184 (all panels). Are these differences significant? If these differences are not (B and E) or are (C and D), it is unclear what this figure adds to the manuscript. Depending on results of the statistical analysis (or if a statistical analysis cannot be performed), the authors should consider removing the figure (and editing in the text for this modification accordingly).

3. Nothing presented in Fig 2, 3, 4, and 5 suggests that some level of RDT cannot be accomplished through Spot 42 region I hybridization (or that region II hybridization alone is sufficient for RDT), just that the short isoform does not accumulate in region II variants. Unless an experiment that determines whether region I/II variants disrupt RDT can be performed, region I hybridization should be depicted in Figure 7A; the possibility of region I contributions to RDT should also be acknowledged in the text.

3. The authors seem to make an assumption (given the data presented in Fig 6) that defective Spot 42 base-pairing (Table 1) leads to an altered (reduced) metabolite flux toward an anabolic pathway (possibly reduced metabolite moving toward UDP-glucose/galactose) through Gal protein fine-tuning. However, such specific data is lacking in the manuscript and would make a substantial contribution to Fig 6 (i.e., western blots of Gal protein levels with ectopic expression of Spot 42 [and region I and II variants, along with associated suppressors]).

Minor revisions:

Ln 31: Use “Rho” in place of “the Rho factor”.

Ln 83: Put commas between Spot 42: “The sRNA, Spot 42,…”

Ln 98: Delete the comma after reference 24.

Lns 100-2: It is unclear what the authors mean by new synthesis not being required for the two different 3’ ends (new synthesis would be required [galET-long cannot be generated from galET-short, and it was not concluded that galET-short can be generated from galET-long]).

Lns 107-10: Please add a reference for Spot 42 remaining maximal in LB (especially w/o glucose) at 0.6.

Lns 134-6: This sentence (“Using 3’ RACE…”) seems like it would be better with the following paragraph (beginning at Ln 137).

Ln 212: This section caption is especially uninformative. Consider changing to: “Both RNase E and Rho serve a function in the mechanisms generating unique galET 3’ ends”

Ln 231: Consider using: “To test whether…” vs. “To test if…” here and elsewhere.

Ln 235: Perhaps it would be better to say “We found that…” vs. “The results showed that…” (less passive).

Ln 237: This can be part of the previous paragraph; the line space is not necessary.

Ln 238: This may be changed to: "In particular, as a functional Rho is present in the rnets mutant grown at a non-permissive temperature, the drastic reduction in galET-long levels indicates Rho does not serve a significant function during the generation of this transcript isoform."

Ln 241: Consider changing "...but this requires RNase E" to "... in an RNase E-dependent manner"

Ln 246: Consider using “The cell-free system contains all…” instead of “The cell-free system is supposed to contain all…”

Ln 273: It seems appropriate to speculate that RNase E may serve some function generating the 2,121 and 2,166 nucleotide RNAs in the text; the BL21(DE3) cells still produce a functional RNase E (vs. the less functional truncated RNase E variant from BL21 Star [DE3]).

Ln 276: Double-check reference formatting (other references are in brackets)

Ln 298: The authors should explain the properties of the rho::AmpR allele. Is the rho nonfunctional? Rho is essential; therefore, this mutation is unlikely to produce a nonfunctional variant of Rho. In what way do the properties of rho::AmpR differ from rho15(Ts)?

Ln 308: This would be a good spot to introduce the supplemental data on exoribonucleases (vs. introducing this particular new data in the discussion). Although, the results are negative, as noted in the manuscript discussion (should also be acknowledged at this line in the text if the figure is added here), these RNases have redundant function (double mutants are not viable). It may be worthwhile to speculate here on the function of RNases toward dsRNA (such as RNase III) for generation of galET-short. The point could be made that some RNase is necessary to generate galET-short (the transcript from RDT alone would produce a transcript that is too long to be this isoform). This point would be parallel to the section beginning at Ln 399 in the discussion.

Ln 313: I believe the line should read: "Here we also found...."

Ln 330: change to “…where both anabolism and catabolism of galactose are required.”

Lns 339-41: Consider editing to: “In exponential growth phase, the doubling times of cells encoding galMMI-1 or galMMII-1 (SI Appendix, S1 Fig B) were 22 and 26 % longer than those encoding the WT gal sequence, respectively (Table 1).”

Ln 386: Although the authors present data suggesting the 5’ phosphorylated state of Spot 42 state does not significantly contribute to RNase E-mediated gal transcript processing, it does not rule out the potential internal entry (5’ bypass) pathway also noted in reference 43 (Bandyra et al. 2012. Mol Cell 47:943-53). As the 5’-dependent pathways seem to have unclear importance (or possibly be redundant to the internal entry pathway) for Spot 42-promoted RNA processing, it seems relevant to note the possibility that this alternative pathway makes a contribution to gal transcript processing in the text.

Ln 775: Delete the comma at the end of the line.

Lns 799-801: The mutation names should match what is shown in the figure (i.e., these can be: “GW20 (amsts [rnets]…”

Ln 806: Colors of the bars should be noted with respective transcript isoforms.

Ln 809: Change the figure panel lettering (and corresponding caption and spots in the text) such that the blots are A-C (look at Ln 813; the ladders are in all three blots, but not B).

Ln 817: I believe this is a sliding window GC-content plot; it is customary to say this is what is being depicted and the window size being used to determine GC-content (in the figure caption and text).

Ln 823: The word “the” should be lowercased. Also, please decide on a consistent choice for capitalization state of “lane” in the caption.

Ln 830: “were taken…” instead of “was taken…”

Ln 834: Technically, the figure does not assay gal expression. The caption should reflect tuning of protein levels toward an anabolic pathway.

Ln 865: The color gray should be lowercased. Also, depending on formatting, these text colors may not be included in the final manuscript version of the manuscript.

Ln 886: Figure S4 is not referenced anywhere in the text. Please make reference to this figure where appropriate.

Ln 935: The font is difficult to read on Fig 6B. It would be more informative to show the individual growth curves (vs. mean and SDs). Do the authors have data points for LB+0.5% galactose (Δspf) beyond 500 minutes?

Ln 958: It would be useful for the authors to show a representation of the assumed cleavage events leading to galET-short in the model (perhaps just by noting “RNase?” in figure panel 7B). It would also be helpful if “RNase E/RNase?” (instead of RNase E alone) cleavages were shown closer at the relevant ends of galET-long and -short (it is unknown precisely where the cleavage/processing events happen, so drawing the cleavage at relevant sites would not be any less inaccurate).

**Have all data underlying the figures and results presented in the manuscript been provided?**

Reviewer #1: Yes

Reviewer #2: Yes

Reviewer #3: None

Reviewer #4: Yes

PLOS authors have the option to publish the peer review history of their article (what does this mean?). If published, this will include your full peer review and any attached files.

Reviewer #1: No

Reviewer #2: **Yes: **Sankar Adhya

Reviewer #3: No

Reviewer #4: No

---

## [Decision Letter · Decision Letter 1]

29 Sep 2021

Dear Dr Lim,

Thank you very much for submitting your Research Article entitled 'sRNA‑mediated regulation of gal mRNA in E. coli: Involvement of transcript cleavage by RNase E together with Rho‑dependent transcription termination' to PLOS Genetics.

The manuscript was fully evaluated at the editorial level and by independent peer reviewers. The reviewers appreciated the attention to an important topic but identified some concerns that we ask you address in a revised manuscript

We therefore ask you to modify the manuscript according to the review recommendations. Your revisions should address the specific points made by each reviewer.

[LINK]

Yours sincerely,

Jue D. Wang

Associate Editor

PLOS Genetics

Josep Casadesús

Section Editor: Prokaryotic Genetics

PLOS Genetics

Reviewer's Responses to Questions

**Comments to the Authors:**

Reviewer #1: The revised manuscript is much improved and answers the criticisms to this reviewer's satisfaction.

Reviewer #2: The authors have satisfactorily revised the manuscript adequately responded to the criticisms of the reviews. I recommend its publication in Plos Genetics.

Reviewer #3: A revised manuscript by Jeon et al. presents some additional data, most notably on the involvement of RNAses in the gal operon regulation, and remove other pieces of data that were questioned by the reviewers. The manuscript is easier to follow and provides some interesting insights into gal control. It is customary, however, to ask if the authors have addressed all the reviewers’ concerns. This is not the case with mine.

I am still not convinced that the data on Rho have any novelty. It is encouraging that Rho terminates transcription when RNAP is paused at a consensus site, since this is consistent with what we know. But there does not seem to be any direct interplay between Spot 42 and Rho. The authors mention a possibility that Spot 42 RNA binding to region 1 could have an effect on termination (incl. in Figure 7) but nothing in their data suggests this. In the first round or review, I wrote: “the authors should inhibit translation of galK by other means to distinguish different aspects of its contribution to gal control”. This would have been a very simple way to test if Spot 42 does anything in addition to inhibiting translation initiation. Since the plasmids are used in the main “extract” assay, a substitution of the galK start codon would take very little effort. In the absence of this, or any other similar evidence, the null hypothesis is that spot42 inhibits translation, just as expected, and this promotes Rho-dependent termination.

Another request that was made by me and other reviewers but was ignored is a need to report error analysis that would indicate that the results are reproducible. I understand that not every reviewer’s comment has to be addressed, as long as a valid reason is offered. I do not, however, think that the lack of evidence in support of reproducibility is acceptable. The figure legends have to state the number of replicates used. If any fold effect is stated (2-fold, 50% etc), it must be accompanied by error analysis. Otherwise, the numbers should be substituted by some qualitative terms.

With respect to 2 different mechanisms of control, what is really happening here is that the inhibition of translation by spot 42 leads to generation of two shorter RNA species, one via premature termination of the nascent RNA (possibly every transcript, if the pause site in question is strong), the other – by ribonuclease processing of RNAs that have been completely made. In both cases, spot 42 job is to make sure that the proximal RNA, and thus proteins, are present in excess. But from the physiological point of view, these two RNAs should be identical, as their ends are closely spaced. So the outcome is actually the same, just the means differ.

I suggest making the following changes to Figure 7:

Panel A: remove region I base pairing from the picture and draw ribosome at a real scale = it protects ~23 nt and depicting this correctly would help to show why base pairing in region II does its job

Panel B: in late 2020, papers showing that Rho travels with RNAP and terminates transcription when the enzyme pauses and RNA is captured were published. They should be cited, and the cartoon should reflect these new data.

Reviewer #4: The authors have done a good job addressing most the reviewer’s comments and have added new results describing how another well-characterized RNase (RNase III) serves a function generating galET-short. This new finding adds to the authors’ story of how different binding events and parts of Spot 42 RNA potentiate different galET processing events. The authors are suitably careful to leave open the possibility that RNase III processing may be the result of dsRNA that forms intramolecularly vs. between gal and Spot 42—this possibility is acknowledged in the model figure. However, two major, related points raised in the initial review remain unaddressed after revision. These unadressed points concern reproducibility of the results in the form of enumerating the number of replicates performed or their statistical analysis. Addressing these points will greatly strengthen the conclusions presented in this manuscript.

Major points

1. The authors need to include information regarding the number of replicates performed for representative blot presented in the manuscript. This need could be met simply by stating in the figure captions: “The data presented in these blots are representative of X independent experiments.”

2. Fig 6B. Error bars are depicted, but it remains unclear from the text and figure legend what these error bars represent. Presumably, these are the same errors that are propagated to determine the errors in the calculated doubling times presented in Table 1. The error analysis must be fully explained. Are they standard deviations of replicate data? Standard errors of means? The authors should also specify “n=x”, where x is the number of replicates.

Minor point

1. In the Figure 7 caption, it would be better to state directly that “RNase(?)” represents currently unidentified RNases(s) that may be involved in the further processing events of the galET transcripts.

**Have all data underlying the figures and results presented in the manuscript been provided?**

Reviewer #1: Yes

Reviewer #2: Yes

Reviewer #3: Yes

Reviewer #4: **No: **see review comments.

PLOS authors have the option to publish the peer review history of their article (what does this mean?). If published, this will include your full peer review and any attached files.

Reviewer #1: **Yes: **max gottesman

Reviewer #2: **Yes: **Sankar Adhya

Reviewer #3: No

Reviewer #4: No

---

## [Decision Letter · Decision Letter 2]

14 Oct 2021

Dear Dr Lim,

We are pleased to inform you that your manuscript entitled "sRNA‑mediated regulation of gal mRNA in E. coli: Involvement of transcript cleavage by RNase E together with Rho‑dependent transcription termination" has been editorially accepted for publication in PLOS Genetics. Congratulations!

Yours sincerely,

Jue D. Wang

Associate Editor

PLOS Genetics

Josep Casadesús

Section Editor: Prokaryotic Genetics

PLOS Genetics

Comments from the reviewers (if applicable):

Reviewer's Responses to Questions

**Comments to the Authors:**

Reviewer #3: The revised manuscript addresses most of my concerns and is suitable for publication.

Reviewer #4: The authors have fixed the remaining small issues with the manuscript.

**Have all data underlying the figures and results presented in the manuscript been provided?**

Reviewer #3: None

Reviewer #4: Yes

PLOS authors have the option to publish the peer review history of their article (what does this mean?). If published, this will include your full peer review and any attached files.

Reviewer #3: No

Reviewer #4: No

**Data Deposition**

http://datadryad.org/submit?journalID=pgenetics&manu=PGENETICS-D-21-00643R2

**Press Queries**

---

## [Editor Report · Acceptance letter]

25 Oct 2021

PGENETICS-D-21-00643R2 

sRNA‑mediated regulation of gal mRNA in E. coli: Involvement of transcript cleavage by RNase E together with Rho‑dependent transcription termination 

Dear Dr Lim, 

We are pleased to inform you that your manuscript entitled "sRNA‑mediated regulation of gal mRNA in E. coli: Involvement of transcript cleavage by RNase E together with Rho‑dependent transcription termination" has been formally accepted for publication in PLOS Genetics! Your manuscript is now with our production department and you will be notified of the publication date in due course.

With kind regards,

Livia Horvath

PLOS Genetics

On behalf of:
